# Assessment of the Effectiveness, Socio-Economic Impact and Implementation of a Digital Solution for Patients with Advanced Chronic Diseases: The ADLIFE Study Protocol

**DOI:** 10.3390/ijerph20043152

**Published:** 2023-02-10

**Authors:** Borja García-Lorenzo, Ania Gorostiza, Nerea González, Igor Larrañaga, Maider Mateo-Abad, Ana Ortega-Gil, Janika Bloemeke, Oliver Groene, Itziar Vergara, Javier Mar, Sarah N. Lim Choi Keung, Theodoros N. Arvanitis, Rachelle Kaye, Elinor Dahary Halevy, Baraka Nahir, Fritz Arndt, Anne Dichmann Sorknæs, Natassia Kamilla Juul, Mikael Lilja, Marie Holm Sherman, Gokce Banu Laleci Erturkmen, Mustafa Yuksel, Tim Robbins, Ioannis Kyrou, Harpal Randeva, Roma Maguire, Lisa McCann, Morven Miller, Margaret Moore, John Connaghan, Ane Fullaondo, Dolores Verdoy, Esteban de Manuel Keenoy

**Affiliations:** 1Kronikgune Institute for Health Services Research, Ronda de Azkue 1, Torre del Bilbao Exhibition Centre, 48902 Barakaldo, Basque Country, Spain; 2Osakidetza Basque Health Service, Barrualde-Galdakao, Integrated Health Organisation, 48960 Galdakao, Spain; 3Biodonostia Health Research Institute, Paseo Dr. Begiristain s/n, 20014 Donostia, Basque Country, Spain; 4OptiMedis, Burchardstrasse 17, 20095 Hamburg, Germany; 5Unidad de Investigación AP-OSIs, Hospital Alto Deba, 20500 Arrasate-Mondragón, Gipuzkoa, Spain; 6Instituto de Investigación Sanitaria Biodonostia, 20014 San Sebastián, Spain; 7Red de Investigación en Servicios de Salud en Enfermedades Crónicas (REDISSEC), 48960 Galdakao, Spain; 8Unidad de Gestión Sanitaria, Hospital Alto Deba, 20500 Arrasate-Mondragón, Gipuzkoa, Spain; 9School of Engineering, University of Birmingham, Birmingham B15 2TT, UK; 10Institute of Digital Healthcare, WMG, University of Warwick, Coventry CV4 7AL, UK; 11Digital & Data Driven Research Unit, University Hospitals Coventry & Warwickshire NHS Trust, Clifford Bridge Road, Coventry CV2 2DX, UK; 12Assuta Medical Centre Ashdod, Ashdod 7747629, Israel; 13Maccabi Healthcare Services Southern Region, Omer 8496500, Israel; 14Gesunder Werra-Meißner-Kreis GmbH, 37269 Eschwege, Germany; 15Internal Medical & Emergency Department M/FAM, OUH, Svendvorg Hospital, Baagøes Allé 15, Indgang 51, 5700 Svendborg, Denmark; 16Department of Public Health and Clinical Medicine, Unit of Research, Education and Development Östersund, Umeå University, 901 87 Umeå, Sweden; 17R&D Project Office, Region Jämtland Härjedalen, 831 30 Östersund, Sweden; 18SRDC, ODTU Teknokent Silikon Blok Kat: 1 No: 16 Cankaya, Ankara 06800, Turkey; 19Department of Computing and Information Sciences, University of Strathclyde, Glasgow G1 1XQ, UK

**Keywords:** digital health, evaluation, mixed-methods approach, effectiveness, socio-economic impact, implementation, chronic obstructive pulmonary disease, heart failure, advanced chronic disease

## Abstract

Due to population ageing and medical advances, people with advanced chronic diseases (ACD) live longer. Such patients are even more likely to face either temporary or permanent reduced functional reserve, which typically further increases their healthcare resource use and the burden of care on their caregiver(s). Accordingly, these patients and their caregiver(s) may benefit from integrated supportive care provided via digitally supported interventions. This approach may either maintain or improve their quality of life, increase their independence, and optimize the healthcare resource use from early stages. ADLIFE is an EU-funded project, aiming to improve the quality of life of older people with ACD by providing integrated personalized care via a digitally enabled toolbox. Indeed, the ADLIFE toolbox is a digital solution which provides patients, caregivers, and health professionals with digitally enabled, integrated, and personalized care, supporting clinical decisions, and encouraging independence and self-management. Here we present the protocol of the ADLIFE study, which is designed to provide robust scientific evidence on the assessment of the effectiveness, socio-economic, implementation, and technology acceptance aspects of the ADLIFE intervention compared to the current standard of care (SoC) when applied in real-life settings of seven different pilot sites across six countries. A quasi-experimental trial following a multicenter, non-randomized, non-concurrent, unblinded, and controlled design will be implemented. Patients in the intervention group will receive the ADLIFE intervention, while patients in the control group will receive SoC. The assessment of the ADLIFE intervention will be conducted using a mixed-methods approach.

## 1. Rationale

Due to population ageing and medical advances, people with chronic disease(s)—including advanced chronic disease(s) (ACDs)—live longer. Patients living with chronic disease may face either temporarily or permanently reduced functionalities, which typically increases their caregiver’s burden and their healthcare resource use [1,2].

Chronic Obstructive Pulmonary Disease (COPD) and Heart Failure (HF) are two of the most prevalent ACDs [3,4]. Both COPD and HF are important causes of morbidity and mortality [3,5] together with high social and economic costs [6,7]. Patients with such conditions may benefit from integrated supportive care provided via digitally supported interventions [8,9]. This approach may either maintain or improve their quality of life, increase their independence, and optimize the healthcare resource use from early stages [1,2]. However, there is limited evidence on the implementation of such approaches.

The EU-funded ADLIFE project (H2020, SC1-DTH-11-2019, 875209) aims to meet the current societal and health system challenges in Europe that are created due to the increasing number of persons living with ACD, often with accompanying co-morbidity and polypharmacy. ADLIFE is designed to provide to such patients and their caregiver(s) with digitally enabled, integrated, and personalized care. The ADLIFE toolbox is a digital solution which consists of three Information and Communication Technology (ICT) components: a platform to manage and customize care plans by the interdisciplinary healthcare team with the patient and for the patient, services supporting the clinical decisions, and a platform encouraging independence and self-management of patients and caregivers. The ICT solutions will be scaled in the seven pilot sites from six countries participating in the intervention in an integrated manner with the existing ICT solutions utilized for healthcare systems.

The ADLIFE intervention objective is to have an impact for three stakeholders: patients, informal caregivers, and healthcare professionals, across the seven participating international healthcare systems. Thus, the ADLIFE intervention aims at slowing down the patients’ functional deterioration ensuring their quality of life and promoting shared decision making, whilst in parallel reducing the caregiver burden, improving the healthcare professional’s working conditions, and optimizing healthcare resource use.

The evaluation of such an intervention is challenging, since multiple dimensions [10,11] arise. Published results have reported notable improvements in some of the primary outcome measures [12,13,14,15], but there is also lack of evidence-based changes in comparison with SoC [16,17,18,19,20]. Therefore, there is a lack of systematic assessments of the effect of personalized integrated care initiatives on patients with ACD and limited consistent evidence on the related impact on conventional and emergency department visits, hospital admissions, and length of hospitalization stay.

The aim of this research is to evaluate whether the ADLIFE intervention, when applied in real-life settings, is able to deliver appropriate targeted and timely care for patients living with ACD. Here we present the protocol of the ADLIFE study. Using a mixed-methods approach, this study will provide robust scientific evidence on the effectiveness, socio-economic implementation, and technology acceptance assessment of ADLIFE compared to the SoC, in order to provide the scientific evidence-based supporting the funding decision-making of the ADLIFE intervention.

## 2. Methods and Analysis

### 2.1. Study Design

The ADLIFE intervention is a quasi-experimental trial following a multicenter, A quasi-experimental, non-randomized, non-concurrent, unblinded, and controlled design. Patients belonging to the intervention group will receive the ADLIFE intervention, while patients in the control group will receive SoC [21]. The SoC description can be found in Appendix A. The trial will be implemented in a real-life setting thus allowing other programs and behavior interventions can be in place. The ADLIFE intervention will be implemented in seven different pilot sites: Basque Country (Osakidetza), United Kingdom (National Health Service Lanarkshire and University Hospitals Coventry and Warwickshire National Health Service Trust), Denmark (Odense University Hospital), Germany (Gesunder Werra-Meißner Kreis), Sweden (Region Jämtland Härjedalen), and Israel (Assuta Ashdod Hospital—Maccabi Healthcare Services Southern Region) involving healthcare professionals, care services, and patients with their caregiver(s). The trial was registered in ClinicalTrials.gov, accessed on October 2022. Number of identification: NCT05575336. The study will be conducted in line with the ethical standards and the applicable European, international, and national law on ethical principles. Ethics approval will be obtained from all pilot sites’ Ethics Committee. Patients and caregivers will provide written informed consent.

#### 2.1.1. Non-Randomized, Non-Concurrent, and Unblinded

The non-randomized design of the ADLIFE clinical trial lies on the recruitment process, as detailed in Section 2.3. To address potential biases and guarantee comparability between the control and the intervention group caused by the afore-mentioned non-random design, the control group selection will be based on a propensity score matching with the intervention group [22]. First, the probability of all eligible individuals of being assigned to the intervention group will be estimated using the patient variables of age and sex, and the number of emergency room (ER) visits and number of hospital admissions observed during the 9 months prior to the ADLIFE intervention. Then, each intervention patient will be matched to a control subject with a similar propensity of being assigned to intervention.

The control group will be retrospectively selected to guarantee that the control group receives usual care. The study will be unblinded, since all stakeholders involved in the intervention group will use the ADLIFE toolbox and will thus be aware of the intervention However, since data will be retrospectively collected from the electronic health records (EHR) for the control and the intervention group, no information bias is expected in the data collection process. The study design and the data collection flow are shown in Figure 1. Data collection is also described in Section 2.7 of this paper.

### 2.2. Study Population

The study population consists of patients with ACD, their informal caregiver(s), and their healthcare professionals, meeting the following eligibility criteria. Eligible patients will have to meet the below inclusion criteria:Aged over 55;HF in functional stage III/IV according to the NYHA scale and/or stages C and D of the ACCF/AHA classification. Stable phase (at least two months without decompensation requiring hospital care);And/or COPD GOLD scale > 2 (FEV1 < 50) and/or mMRC ≥ 2 and/or CAT ≥ 10 and/or use of oxygen at home;With or without comorbidities;They are able to provide informed consent;They still live and generally plan on living in their home for the intervention duration;They or their informal caregiver(s) are able to use digital technology, communication tools, and/or networks and have access to a computer, laptop, tablet or smartphone and wifi/internet connection;They or their informal caregiver(s) understand, read, and talk the native language.

The informal caregiver will be a person who provides occasional or regular support to the patient needs. Caregivers will be eligible if the patient they care for meets the inclusion criteria and is included in the study. Healthcare professionals will be eligible if they are involved in the included patients’ care, are open to new ways of working, specifically as part of a coordinated and collaborative team, and are also open to the use of new technology. Criteria for patient exclusion include an existing diagnosis of an active malignant neoplastic disease, being in any active list of transplantation, or refusing/unable to sign the informed consent. Patients who have withdrawn their participation from ADLIFE will be excluded; caregivers will not be eligible if the patients they care for meet the exclusion criteria; Healthcare professionals not caring for eligible patients will not be included.

### 2.3. Recruitment

The recruitment of healthcare professional, patients and caregivers will last 7 months. Healthcare professionals will be recruited during the first 4 months by each pilot site using a convenience sampling based on their individual profiles. The healthcare professionals will be contacted via email, letter, or face-to-face meeting to provide them with initial information about the ADLIFE intervention. Healthcare professionals consenting to participate will be recruited.

Once healthcare professionals have been recruited and trained in the use of the ADLIFE intervention, each pilot site will identify a tentative target patient population fulfilling the eligibility criteria from EHR. After a validation process conducted by the healthcare professionals, a final target population will be defined.

The intervention and control group will be recruited from the final target population during the last 2 months. Candidates for the intervention group will be recruited by healthcare professionals or research assistants according to the inclusion/exclusion criteria. They will be contacted by email, post, phone, or face-to-face meetings, and provided with information describing the ADLIFE intervention, their expected role, and their required ICT skills. Patients participating in the intervention will sign an informed consent and will receive ADLIFE intervention training.

Patients in the intervention group may appoint an informal caregiver who will participate with them in ADLIFE in order to assist them with ICT management and self-management empowerment of their disease. Informal caregivers will sign an informed consent to participate in the intervention, and a second informed consent to access their corresponding patient’s data. They will receive ADLIFE training together with the corresponding recruited patient they care for.

In the event of a participant’s drop-out of the project, their previously collected data will be retained, unless otherwise stated, and analyzed under the intention-to-treat principle. Drop-out reasons will be defined as: (i) death, (ii) not interested in the intervention anymore, (iii) too time-consuming, (iv) technology issues, (v) declaring lack of help from their informal caregiver, (vi) institutionalization, which implies a change of healthcare professional, and (vii) no response. In the case of healthcare professionals, any position change will also be considered as drop-out reason. Informal caregivers will automatically drop-out of the project if their patients do so.

After removing the intervention patients from the final target population, a target control population will be identified. Control group patients will be retrospectively selected based on the afore-mentioned matching techniques (see Section 2.1.1.) from the target control population. No signed informed consent will be needed since anonymous data will be extracted from EHR by the pilot sites. Intervention patients withdrawing from the intervention will not be eligible for the control group. The recruitment process is described in Figure 2.

### 2.4. The ADLIFE Intervention and Standard of Care

The ADLIFE intervention consists of the deployment and use of the ADLIFE toolbox by patients, informal caregivers, and healthcare professionals in the afore-mentioned pilot settings.

The ADLIFE toolbox involves two interconnected platforms. Patients will use the Patient Empowerment Platform (PEP), and healthcare professionals will be assisted by clinical decision support services within the Personalized Care Plan Management Platform (PCPMP). Patients participating in ADLIFE will have a personalized care plan, created in PCPMP, which will be developed and managed together with their healthcare professionals. PCPMP will be used in integration with the clinical sites’ ICT systems to create patient care plans based on each patient’s baseline and most recent clinical data, following clinical evidence. PEP will facilitate the patient’s independence and self-management by presenting their personalized goals, activities, and educational materials, collecting their observations and questionnaires, and providing real-time interventions tailored to the patient’s lifestyle.

The main task of patients and their informal caregivers will be to use the PEP as part of their healthcare management process together with their healthcare professionals. The ADLIFE intervention will consider the health-related outcomes relevant for the patient in actual health service planning and evaluation. By identifying the outcome that will be responsive to each measure, professionals and patients will have the chance of reviewing the health-related outcomes, and of jointly choosing the activity, objective or goal that boosts the desired one. The health-related outcomes will be reflected as labels that bind every activity, goal, and/or indicator included in a care plan. The labelling mechanism has been co-created with healthcare professionals and automated to enable health-related outcome tracking over time and over a wide spectrum of patients.

The control group follows the SoC according to the pilot site organizations’ criteria. Since pilot sites belong to different health care systems, information on SoC was derived from semi-structured interviews with three stakeholder groups, 5–7 persons in each group [21].

### 2.5. Outcomes

The set of health-related outcomes grouped around relevant domains for the ADLIFE target population was defined following the International Consortium for Health Outcomes Measurement (ICHOM) methodology (see online the Appendix A for details) [23,24]. The health-related outcome set provides a consensual definition of desired end results for both the outcome-based care planning and for the project evaluation. The framework groups each of the primary and secondary outcomes by the broad health-related outcome to which it is responsive. Figure 3 shows the Health-Related Outcome set for people over 55 years old with severe heart failure and/or COPD. The conceptual data framework identifies the relevant health areas (outer ring).Each health outcome area comprehends multiple dimensions (inner ring).

Besides health-related outcomes, implementation-related outcomes including barriers/facilitators related to the implementation process, technology acceptance and adoption, and contextual factors for further exploitation to later scaling-up will also be assessed. To define the variables of interest, a framework for implementation assessment was developed (see Figure 4) which is based on a human, organization, and technology-fit (HOT-fit) framework as a Health Information System evaluation framework [25], and is complemented with the intervention, process, and outer setting of the Consolidated Framework for Implementation Research (CFIR) [26]. Together, these frameworks provide a structured and systematic way to identify constructs influencing the implementation of ADLIFE on different levels.

#### 2.5.1. Primary

The primary outcome will be the number of ER visits during the 9-month follow-up. The ER visits regularly become a crucial point in healthcare systems when care is poorly coordinated. A high number of ER visits has been associated with functional decline and mortality [27]. In this context, existing evidence suggests that frequent ED users are at-risk patients for whom interventions may improve health outcomes [28,29]. Then, this primary outcome, defined as the main effectiveness outcome, is a proxy of the appropriateness of care in real-life settings for patients with ADC, and consequently, a gain in health-related outcomes [27,28,29]. ER visits will be collected from the EHR.

#### 2.5.2. Secondary

The secondary outcomes will be assessed on patients, caregivers, and healthcare professionals.

Patients will be assessed on:Patient-Reported Outcome Measurements (PROMs):Health-related quality of life (EQ-5D-5L) [30];Mood/emotional health (HADS—Hospital Anxiety and Depression Scale) [31];Activities of daily living (Lawton scale, Barthel Index, Kansas City Cardiomyopathy Questionnaire score, COPD assessment test score) [32,33,34,35];Complexity (Modified Medical Research Council—mMRC—Dyspnea Scale) [36].
Technology acceptance and future adoption of the ADLIFE intervention (the Unified Theory of Acceptance and Use of Technology, UTAUT) [37];Resource use and their associated costs will be also assessed on patients [38] (see data collection guide 2 in Appendix A);Caregiver will be assessed on:Burden (Zarit Burden Interview, ZBI) [39];Mental well-being (Warwick-Edinburgh Mental Wellbeing Scale, WEMWBS) [40].Healthcare professionals will be qualitatively assessed on:Perceived coordination among settings;Quality of the integration of care;Decision making process;Working conditions.All stakeholders will be qualitatively assessed on:Perceived communication;Satisfaction with accessibility, security, and personalized care plans;Barriers/facilitators related to the implementation process.

Within the scope of the implementation assessment, contextual factors focusing on local technological, organizational, and human factors, will also be qualitatively assessed before and one year after the start of the intervention in a sub-group of different stakeholders including physicians, general practitioners, nurses, health coordinators, IT staff, and hospital CEOs [41].

### 2.6. Sample Size

The sample size was calculated using the number of ER visits as the main primary outcome. In order to detect an effect size of 0.6 ER visits per year, with a standard deviation of 1.2 with a 5% level of significance, a 90% of statistical power set, assuming a conservative intra-cluster correlation coefficient of 0.06 (each pilot site defined as a cluster) and a drop-out rate of 30%, 1692 patients will be required (846 per branch) across pilot sites.

For the qualitative approach, a purposeful sampling will be used to select the participants, recruiting those participants who might provide in-depth and detailed information about the ADLIFE intervention. The pre-intervention interviews for the implementation assessment will involve: one medical director, six healthcare professionals (two physicians, two general practitioners, and two nurses), and two IT staff for each pilot site. The post-intervention interviews will be structured in the same way, adding three to six patients and three to six informal caregivers for each site.

### 2.7. Data Collection

A data collection guide including a codebook and a template will be provided to pilot sites to conduct the quantitative data collection on socio-demographic, clinical, economic, and usability variables. Variables will be observed at baseline and at endpoint, except for resource use which will be measured during the 9-months before and after the baseline, and except for the usability variables which will be measured at endline. Data collection flows are shown in Figure 1. Data will be collected from three data sources: the EHR, Qualtrics questionnaire for technology acceptance completed by participants, and the on FHIR repository [42]. For the effectiveness and socio-economic evaluation, each pilot site will provide its corresponding dataset to the evaluator site to be merged into a single data space. Data collection guides 1 and 2 in the Appendix A show comprehensive variable lists to be collected. For the technology acceptance and adoption, pilot sites will communicate the online questionnaire links to the participants and the data will be collected. A data collection guide for the technology acceptance and adoption evaluation can be found in the Appendix A.

For the qualitative work, semi-structured interviews will be conducted on patients, informal caregivers, healthcare professionals, managers, and IT staff. A detailed protocol with the data collection guide and templates to be used for summarizing the data interviews will be circulated across pilot sites. Interviews at each pilot site will be conducted in the corresponding national language, and will preferably take place face-to-face, or virtually if necessary (telephone or videoconference). The interviews will be audio recorded and verbally transcribed to a structured template. For effectiveness assessment, data from interviews will be collected after follow-up period, whereas for implementation assessment a pre-post approach will be followed.

### 2.8. Data Management

A robust approach to data protection and data management is adopted prior to any contact with patients or their health data. This approach is described in the Data Management Plan (DMP) [43]. The DMP describes the data protection roadmap for each of the five categories of patient level data (mock healthcare data, training healthcare data, control healthcare data, intervention healthcare data and patient/HCP reported data). At a glance, the DMP presents the data processing and flows that will largely take place within each of the seven pilot sites, to identify patients who match the eligibility criteria for the ADLIFE intervention, and then how these data will be processed including pseudonymization and anonymization steps, before being made available for wider consortium use. This wider use includes the design and development of technical components, and the training and validation of artificial intelligence algorithms. The anonymization process begins with the classification of the sensitivity of those variables that are essential for the purpose of the study. The protection criteria established include the anonymization of direct identifiers using hash algorithms, and the anonymization of indirect identifiers using micro-aggregation and temporal perturbation, reducing the risk of re-identification of the information as a whole to 5%. It will therefore be ensured that, within reasonableness and within the procedures adopted, no personal data or datasets can be associated to any person, within a reasonable time.

The DMP also focuses on the formal Data Management Plan template published by Horizon 2020. This mostly confirms the intention of the project to make available some open research data at the end of the project, and how it intends to comply with the FAIR principles. The DMP therefore, sets the principles to enable a privacy-by-design approach to data sharing by balancing the benefits of using personal health data with a range of risk-management controls.

### 2.9. Analysis

The ADLIFE study will provide the assessment of the effectiveness, socio-economic, implementation, and technology acceptance aspects of the ADLIFE intervention compared to the SoC. The threefold approach is presented in the following subsections.

#### 2.9.1. Effectiveness Assessment

A mixed assessment strategy will be performed using quantitative and qualitative methods.

For the quantitative approach, first a descriptive analysis followed by univariate statistical tests will be conducted. The effect of the intervention will then be assessed by generalized mixed models for longitudinal data. Linear models will be used for continuous variables and logistic models for dichotomous variables. Given the hierarchical structure of the data where patients are nested in pilot sites, the latter will be included as random effect to control the models by the variability across sites. In order to consider the different time of follow-up of each participant, all models will be adjusted by this factor, i.e., the time of follow-up will be included as a covariable. All models will also be adjusted for potentially confounding factors. Since healthcare services data are usually characterized by being discrete, zero-inflated counts and right-skewed, special attention will be paid to the selection of the distribution which best fits the data. In all quantitative analyses, we will use an intention-to-treat approach and set the level of significance at *p* < 0.05.

For the qualitative approach, a two-step analysis will be conducted. The pilot sites will first perform a content analysis in their own languages [44], based on a systematic approach to determine trends and patterns within the text, and will organize and identify topics and their relationships. For each identified topic, a comprehensive set of quotes will be transcribed to ensure comparability across regions. Then, in a second step, an aggregate cross-country analysis to assess the validity of the results, by comparing the main results among the pilot sites and the different stakeholders, will be conducted by the evaluation coordinator. The COREQ Checklist (Consolidated criteria for Reporting Qualitative research) [45] for quality assurance will be followed.

#### 2.9.2. Implementation Assessment

For the quantitative approach technology acceptance and adoption assessment, the questionnaire data collected at two timepoints in the intervention, will be exported from Qualtrics to a statistical software package [46]. Descriptive statistics will be used to summarize the participants’ demographics and core set of UTAUT constructs. To measure the reliability of the UTAUT model’s constructs and form correlations between them, data analysis will be performed using techniques such as structural equation modelling, a multivariate statistical analysis technique that is used to analyze structural relationships and tests the underlying factors and hypotheses.

For the qualitative analysis of the contextual factors a similar two-step analysis approach is planned, as described in in the effectiveness assessment in Section 1. This includes a content analysis in the national language using a standardized coding system based on the HOT-fit framework, followed by an aggregated cross-country analysis in English to summarize the main content of all pilot sites according to the HOT-fit framework and support it with respective quotes. The COREQ Checklist [45] for quality assurance will be followed.

#### 2.9.3. Socio-Economic Assessment

Simulation modelling applying discrete event simulation (DES) [47] technique will be used to represent the natural history of patients with ACD and estimate the long-term socio-economic impact. A common simulation model will be estimated for all pilot sites and then, adapted to each pilot site situation. First, the conceptual model that will rule the simulation model will be agreed and defined with experts. Second, the simulation model will be built up to represent the SoC scenario and validated by contrasting it with the data observed in real life. After being calibrated and validated, in the third step the previously estimated ADLIFE effectiveness will be added to the model in order to also represent the ADLIFE scenario. Finally, for both scenarios, the budget impact analysis (BIA) will be estimated multiplying the resource consumption rate by unit costs, and then forecasted using population projection. Then, the burden of the disease under both scenarios will be obtained over time and the changes in the expenditure of the healthcare system after the adoption of the intervention addressed.

## 3. Discussion

This study will conduct a holistic evaluation, not only based on a mixed-methods approach addressing the effectiveness, technology acceptance, and socio-economic impact, but also by involving the key stakeholders: patients, caregivers, and healthcare professionals. In addition, the use of the health outcome framework will shape the scientific evidence compared to the SoC. Study findings will be disseminated through peer-reviewed publication and scientific conferences [48].

However, this study is not free of limitations. The first is related to the recruitment process. The healthcare professionals/research assistants will recruit the intervention participants according to their subjective assessment, which will not allow for the participants’ randomization. Second, the design of the data collection process, where the control period information will be retrospectively collected, will not allow the quality of life to be used as a main outcome. Moreover, thirdly, the existing differences in the ADLIFE intervention and the SoC across the different health systems might lead to a bias of the ADLIFE global effect. To address this twofold concern, the ADLIFE intervention must first be defined as a pragmatic trial [49] of a complex intervention [50]. Pragmatic trials measure effectiveness in routine clinical practice and its design reflects variations across patients. While interventions should be precisely described in Randomized Control Trials (RCT), in pragmatic trials this does not imply that the same interventions are offered to each patient. Indeed, clinicians and patients’ biases are not necessarily considered as a weakness but accepted as part of the physicians and patients’ responses to the intervention, and therefore included in the overall assessment [51]. Pragmatic trials may vary across similar participants, by chance, by practitioner preference, and according to institutional policies.

Furthermore, the ADLIFE intervention as a complex intervention implies working closely with local stakeholders, and considering the implementation as an iterative, recursive, and long-term process. The complex interventions allow the implementation to vary across different contexts with no loss of the core intervention components. The standardization of interventions might be rather into the underlying process and functions than on the specific form of the components delivered [51]. In the context of the ADLIFE intervention, the differences across healthcare systems becomes a challenge, but also it becomes a strength when assessing the intervention across diverse clinical real settings. Moreover, the methodological approach offers the chance to control for the afore-mentioned differences when estimating the ADLIFE intervention effect, including the site as random effect for this purpose.

## Figures and Tables

**Figure 1 ijerph-20-03152-f001:**
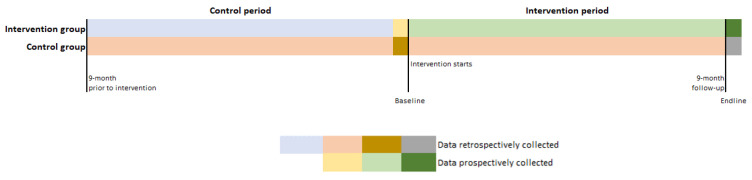
Study design.

**Figure 2 ijerph-20-03152-f002:**
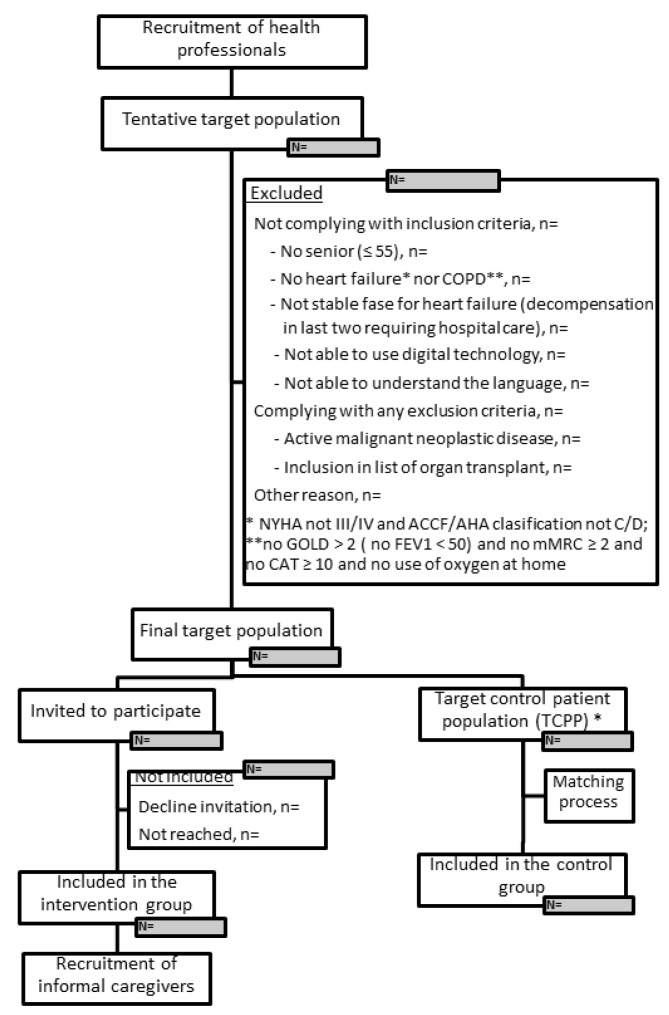
Flow-chart of the recruitment and selection process. * TCPP is defined as the subset of final target population after removing the intervention patients.

**Figure 3 ijerph-20-03152-f003:**
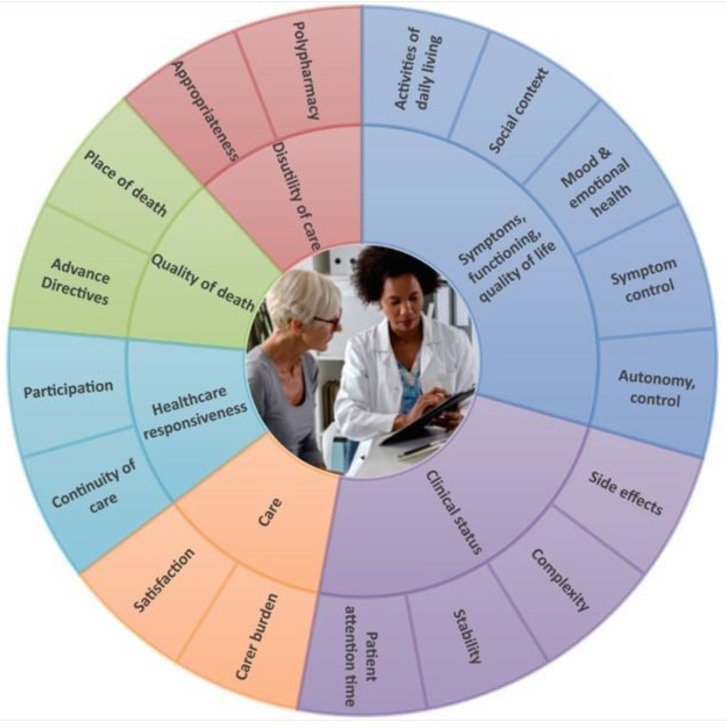
Health-related outcome set for patients with Advanced Chronic Disease. Adapted from ICHOM.

**Figure 4 ijerph-20-03152-f004:**
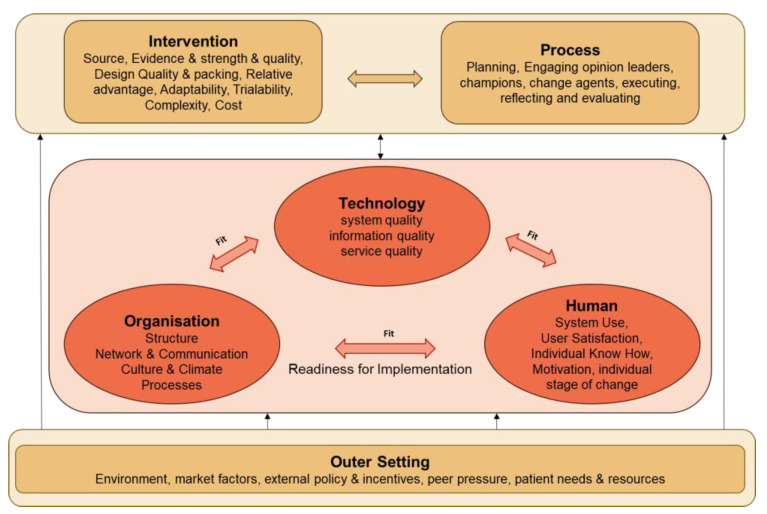
ADLIFE framework for implementation assessment.

## Data Availability

Not applicable.

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
