# Peer review of "Assessment of the Effectiveness, Socio-Economic Impact and Implementation of a Digital Solution for Patients with Advanced Chronic Diseases: The ADLIFE Study Protocol"

_ijerph, 2023, doi:10.3390/ijerph20043152_

Round 1
Reviewer 1 Report
This is an interesting and well-designed research protocol. I wish the content of the protocol a successful implementation. I'd like to leave you with a few comments in this regard.
1) How do you plan to solve the problem of uniformity of interventions across the 7 pilot test sites?
2) [line 129] It would be good if an explanation for a propensitiy score was added.
3) [line 180] More specific comments is needed on what the patient recruitment period will be like.
4) [line 213] How can you eusure anonymity?
5) [line 318] Is it appropriate to set the ER visit as the primary outcome?
Reviewer 2 Report
Thank you for the opportunity to review this article entitled "Assessment of the effectiveness, socio-economic impact and implementation of a digital solution for patients with advanced chronic diseases: The ADLIFE study protocol"
Overall, I believe this study serve as a body of literature regarding this area of study regarding the QoL of patients with advanced diseases, using ED visit as a surrogate variable. However, the methods are not clear and largely needed to be edited.
1. The authors stated that they aimed to evaluate whether ADLIFE intervention is able to deliver appropriate care. I'm afraid that by using this non-randomized study between ADLIFE intervention and standard of care (SoC) would address this issue. Patients in ADLIFE group are totally received the different intervention (compared to the current intervention) which may lead to some selection bias.
2. Also, the authors will selected a control group retrospectively from the propensity score based on age, sex, ER visit, and hospital admission. Then, they will use an ER visit as a primary outcome? I can't following this reason. Please clarify.
3. Since this is a multi-center study, a control group would be selected from all sites, making it "heterogenous." Also, the SoC protocol are supposed to be different across all study sites? Please explain more in details about the SoC.
4. I could not understand section 2.5 outcomes. The primary outcome stated is the number of ER visits which will be presented as an average number of ER visits. I believe this action is not a standardized action when evaluating this variable. It should be the total number of ER visits in a pre-specified period?
5. Please explain the first paragraph in the section 2.5. Why do you want to explain this? It's hard for the readers to follow this section like this.
6. I am not a statisticians, but the section 2.9 analysis is too confused to follow. Please consider revise it.
Minor points
7. Please revise figure 1 or consider remove it
8. The authors mention "section c/line 126, section 2/line 212" Please consider revise or edit
9. Line 123: Ref [48] is probably a typo error.
